# Peripheral Nerve Impairment in a Mouse Model of Alzheimer’s Disease

**DOI:** 10.3390/brainsci11091245

**Published:** 2021-09-20

**Authors:** Alessio Torcinaro, Valentina Ricci, Georgios Strimpakos, Francesca De Santa, Silvia Middei

**Affiliations:** 1Institute of Biochemistry and Cell Biology (IBBC), National Research Council of Italy (CNR), Via E. Ramarini, 00015 Rome, Italy; alessio.torcinaro@ibbc.cnr.it (A.T.); valentina_ricci_@hotmail.it (V.R.); georgios.strimpakos@cnr.it (G.S.); francesca.desanta@cnt.it (F.D.S.); 2European Brain Research Institute (EBRI), Viale Regina Elena 295, 00161 Rome, Italy

**Keywords:** Alzheimer’s disease, sarcopenia, skeletal muscle, neurofilament, cholinergic innervation, choline acetyltransferase

## Abstract

Sarcopenia, a geriatric syndrome involving loss of muscle mass and strength, is often associated with the early phases of Alzheimer’s disease (AD). Pathological hallmarks of AD including amyloid β (Aβ) aggregates which can be found in peripheral tissues such as skeletal muscle. However, not much is currently known about their possible involvement in sarcopenia. We investigated neuronal innervation in skeletal muscle of Tg2576 mice, a genetic model for Aβ accumulation. We examined cholinergic innervation of skeletal muscle in adult Tg2576 and wild type mice by immunofluorescence labeling of tibialis anterior (TA) muscle sections using antibodies raised against neurofilament light chain (NFL) and acetylcholine (ACh) synthesizing enzyme choline acetyltransferase (ChAT). Combining this histological approach with real time quantification of mRNA levels of nicotinic acetylcholine receptors, we demonstrated that in the TA of Tg2576 mice, neuronal innervation is significantly reduced and synaptic area is smaller and displays less ChAT content when compared to wild type mice. Our study provides the first evidence of reduced cholinergic innervation of skeletal muscle in a mouse model of Aβ accumulation. This evidence sustains the possibility that sarcopenia in AD originates from Aβ-mediated cholinergic loss.

## 1. Introduction

Alzheimer’s disease (AD) is the most common neurodegenerative pathology affecting ageing. Clinical signs of AD include progressive cognitive decline and memory loss, along with sarcopenia, which is a geriatric syndrome encompassing loss of muscle mass and strength [1,2,3].

Cognitive decline in AD patients commonly leads to their reduced interest toward the surrounding environment, and therefore physical activity and general locomotor activity drop dramatically in these individuals and likely contribute to the development of sarcopenia. However, cumulating evidence indicates that pathological AD markers are also present in skeletal muscle, thereby driving the hypothesis that sarcopenia may share common pathological mechanism(s) with cognitive loss in AD subjects [4]. Since sarcopenia is commonly associated with fast clinical progression of AD pathology [2,5,6], the investigation of its underlying mechanisms has important clinical implications for the identification of novel biomarkers for AD diagnosis and prognosis.

Among the neuropathological events in AD, plaque aggregates of amyloid β (Aβ) produced from proteolytic cleavage of the Amyloid Precursor Protein (APP) are associated with synaptic dysfunction as well as neuronal shrinkage and loss within specific regions of the AD brain [7]. In post mortem brains of AD patients, accumulation of Aβ plaques has been found to have a strong inverse correlation with the expression of pre- and post-synaptic markers of cholinergic activity [8] and with the activity of acetylcholine (ACh) synthesizing enzyme choline acetyltransferase (ChAT) [9], thereby indicating that Aβ aggregates may have a causative role in cholinergic degeneration.

ACh is also highly expressed in the peripheral nervous system, where it regulates autonomic and skeletal muscle functions. In this peripheral system, ACh is released by preganglionic neurons at both sympathetic and parasympathetic nerve fibers and serves all the organs innervated by postganglionic parasympathetic neurons. Furthermore, ACh is the crucial neurotransmitter at the neuromuscular junction (NMJ) between the motor nerve and the skeletal muscle, where it acts to activate muscle contraction [10].

Products of altered APP processing and Aβ accumulation are present in peripheral tissues such as the skeletal muscles [11,12,13,14,15], where aggregated forms of Aβ interfere with ACh release at NMJ [16,17] and with ChAT activity [18]. This evidence supports the possibility that Aβ-related cholinergic dysfunction could be extended to skeletal muscle. However, ACh function in peripheral systems of AD subjects has received little scientific interest so far and the analysis of ACh innervation in skeletal muscle is still missing.

In this study, we investigated the cholinergic phenotype in the skeletal muscle of the AD mouse model Tg2576 [19]. These mice over-express Aβ and have been shown to display signs of sarcopenia, including short and narrow strides and altered duty cycle, when compared to controls [20,21]. We compared cholinergic innervation of the tibialis anterior (TA) between aged Tg2576 mice and their wild type littermates. Our results indicate that Tg2576 mice display reduced neuronal innervation and smaller presynaptic terminals in the TA compared to wild type mice. Furthermore, at the presynaptic level, ChAT content is reduced in Tg2576 mice with respect to their wild type counterparts and this synaptic alteration is accompanied by reduced mRNA levels of Nicotinic ACh receptor, subunit α1 (Chrna1).

## 2. Materials and Methods

### 2.1. Ethical Approval

All tissues used in this study were collected from animals already sacrificed for other experiments, in line with the principle of the three Rs as approved by the Italian Ministry of Health and in accordance with the European Union Directive of 22 September 2010 (2010/63/EU). Heterozygous Tg2576 female mice (APPSWE [Tg2576]), expressing a transgene coding for the 695-amino acid isoform of human Alzheimer β-amyloid (Aβ) precursor protein (Taconic Labs, Rensselaer, New York, NY, USA) carrying the Swedish mutation (APPSwe KM670/671NL mutation; [19]) on a C57BL6/SJL genetic background and their wild type littermates were used for experiments.

### 2.2. Animal Housing and Muscle Harvesting

The animals were housed at IBBC and EBRI conventional animal cores, group-housed (4 mice/cage) with temperature (22 ± 1 °C) and humidity (60 ± 5%) controlled, under a 12 h/12 h light/dark cycle. Animals were fed *ad libitum*.

Six-months-old Tg2576 and their wild type littermate mice were euthanized for tissue harvesting. Briefly, mice were carefully dissected and the tibialis anterior (TA) muscles were embedded in Tissue-Tek OCT compound (Sakura), frozen in liquid nitrogen-cooled isopentane (2-Methylbutane; Sigma-Aldrich, Merk KGaA, Burlington, MA, USA) and stored in a −80 °C freezer for histological analysis; alternatively, TA muscles were snap-frozen in liquid nitrogen for RNA extraction, as previously described [22].

### 2.3. Immunofluorescence of Muscle Sections

For histological analysis, OCT-embedded TA muscles from five wild type and 5 Tg2576 mice were longitudinally sectioned at 20 µm of thickness using a Leica cryostat (Leica CM1850UV) set at −20 °C. Muscle cryosections were thawed, fixed in ice-cold acetone for one minute, air-dried and washed: PBS was used for all washing steps. A blocking solution, containing 4% BSA + 0.1% Triton in PBS, was then added to the sections for 45 min. A primary antibody mix was then added on cryosections and incubated overnight at 4 °C. Muscle sections were then washed three times in 1% BSA and incubated with a mix of secondary antibodies for 45 min at room temperature. Nuclei were counterstained with 4′, 6 diamidino-2-phenylindole (DAPI, Sigma-Aldrich), washed and then mounted using 80% glycerol (Sigma-Aldrich) in PBS and cover slides. Concerning immunofluorescence for the β-amyloid peptide, we followed the same procedure but without using 0.1% Triton in blocking solution.

Primary antibodies used for immunofluorescence were mouse monoclonal antibody raised against Neurofilament light chain (NFL; Santa Cruz Biotechnology, Dallas, TX, USA; Cat# sc-20012; 1:200); goat polyclonal antibody raised against Choline acetyltransferase (ChAT; Millipore, Merk KGaA, Burlington, MA, USA; Cat# AB144P; 1:500); rabbit monoclonal antibody raised against β-amyloid peptide (Aβ, D54D2 XP; Cell Signaling^®^ Technology, Inc. (CST), Danvers, MA, USA; Cat# 8243; 1:500). Secondary antibodies for immunofluorescence were Cy™2 AffiniPure donkey anti-mouse IgG (H+L; Jackson ImmunoResearch, Cambridge, UK; Cat# 715-225-150; 1:300) and Cy™3 AffiniPure donkey anti-mouse IgG (H+L; Cat# 705-165-147; Jackson ImmunoResearch, Cambridge, UK; 1:300); Alexa Fluor^®^ 594 Goat anti-rabbit (H+L; Life Technologies, Thermo Fisher Scientific, Waltham, MA, USA; Cat# A-11012; 1:500).

### 2.4. Image Acquisition

Immunofluorescence of stained TA muscles was acquired using an Olympus confocal microscope (Olympus FV1200) with 40× magnification and visualized with FV10-ASW software (Version 4.2, Olympus, Tokyo, Japan). To reconstruct the longitudinal muscle section entirely, we collected 1024 × 1024 single stack images of NFL signal covering TA muscle section (20 µm of thickness).

For neurofilament analysis, we selected ROIs containing NFL-positive filaments and collected a fixed number of 10 stacks with a Z-step size ranging between 0.15 and 1 µm. For ChAT and NFL analysis, we selected ROIs containing NFL- and ChAT-positive signals and collected 8 to 12 Z-stacks with Z-step sizes of 1 µm.

Aβ staining was acquired using a TCS SP5 confocal laser scanning microscope (Leica Microsystems, Wetzlar, Germany) with 20× magnification and 2× zoom. Specifically, we acquired a fixed number of 5 stacks with a Z-step of 1 µm.

### 2.5. Image Analysis

Quantification was performed by using the ImageJ software (version 1.53c; National Institutes of Health, Bethesda, MD, USA; ImageJ. Available online: http://imagej.nih.gov/ij; accessed on 28 July 2020). Briefly, specific data from Z-stack images were highlighted by creating a Z-project and applying average intensity projection method. RGB images were split to the respective red, green and blue image components. Where necessary, red (ChAT) and green (NFL) channels were merged to a new composite image. The double fluorescence images were thresholded from 0–134 (red) and 10–134 (green) hue values, in order to select and obtain the measurement of the area for each color.

For neurofilament length analysis, NFL-positive filaments were selected according to the following parameters: (1) minimum length of 10 μm, (2) maximum width of 1 μm, and (3) a minimum signal intensity of 100 px. Measurement of segment length was performed manually with segmented line tool. Length values for each data point were added up in order to obtain the total length for each slice. Slice values were then averaged for animal and compared among genotypes.

For synaptic area analysis, elliptical ROIs were drawn in order to select areas where both red and green signals were present. Green (NFL) images were color thresholded with “YEN” algorithm and “YUV” as color space method from 50–170 hue values, to obtain local green area measures for each ROI. Data points from each ROI were considered outliers and excluded from the analysis if they were 1.5 * IQR above the third quartile or below the first quartile. We included in the analysis a total of 106 data points for the wild type group and 93 for the Tg2576 group. Data points were averaged for slice, and then values from each slice were averaged for animal and compared among genotypes.

On ROIs selected for the green channel as described above, red (ChAT) and green (NFL) channels were merged to a new composite image and the double fluorescence images were color thresholded with “YEN” algorithm and “HSB” color space method from 22–47 (orange) hue values, in order to select and obtain the measurement of local orange area for each ROI. Data points were averaged for slice, then values from each slice were averaged for animal and compared among genotypes.

### 2.6. RNA Extraction and qRT-PCR

Liquid-nitrogen frozen TA muscles isolated from six months old female wild type (*n* = 5) and Tg2576 mice (*n* = 6), were homogenized using a tissue homogenizer in Trizol (Sigma-Aldrich). RNA was extracted following Trizol manufacturer’s protocol and then quantified with a NanoDrop instrument. Total RNA was retro-transcribed using random primers and Reverse transcriptase (RT) kit (#N8080234; Thermo Fisher Scientific, Life Technologies, Monza, Italy). qRT-PCR analysis was performed by using 2X SYBR Green Master Mix (Applied Biosystems; Life Technologies, Monza, Italy). The primer pairs were designed by Primer3 website (Primer3 Input. Available online: http://primer3.ut.ee/, accessed on 2 September 2021); the sequences are exon spanning, unless otherwise indicated, and are listed in Appendix A. Quantitative RT-PCR reactions were run on 7900HTABI prism PCR machine. The final analysis of output values was performed using standard ΔΔCt method. The expression level of selected target genes has been measured using TBP gene as housekeeping gene.

For Chrna1 analysis, one data point from the Tg2576 group was considered outlier and excluded from the analysis as it was 2 * SD above the group average.

### 2.7. Statistical Analysis

Data are presented as means plus/minus standard error of the mean (SEM). Statistical differences between groups were verified by Student’s *t*-test (2-tailed). *p* values < 0.05 was considered statistically significant. Cumulative frequencies were compared using Kolmogorov-Smirnov Test. Graphs were generated using GraphPad Prism (GraphPad Software, San Diego, CA, USA).

## 3. Results

### 3.1. Tg2576 Mice Display Reduced Neuritic Length as Compared to Littermate Wild Type

Progressive neuronal shrinkage is a typical hallmark in an Alzheimer’s disease (AD) brain and has been associated with Aβ accumulation [23]. To investigate whether neuritic loss is also evident in neurons of the peripheral nervous system of AD subjects, we measured the length of neuronal segments from the Tibialis Anterior (TA) of Tg2576 and wild type mice. We selected 6-month-old mice as previous studies from our and other groups demonstrated that AD-like pathology is already developed in Tg2576 mice at this age point (o solo age, o time point) [24,25]. Consistently, by using an antibody designed to detect transgenically expressed human APP, we found positive Aβ signal in longitudinal TA sections from Tg2576 but not wild type mice (Appendix A).

In order to label skeletal muscle neurons, we stained longitudinal TA sections with an antibody raised against neurofilament light chain (NFL), a protein expressed in the axons of CNS and PNS neurons (Figure 1A). We then measured the length of NFL-positive filaments in confocal images randomly selected from TA sections, shown in Figure 1B. Images and graphs in Figure 1B,C indicate that the average length of NFL-positive neurites is shorter in TA sections from Tg2576 mice compared to wild type controls (t_(8)_ = 2.683, *p* = 0.0278, Figure 1C). This data implies that axonal shrinkage typical of the AD brain is also evident in the skeletal muscle of Aβ-bearing mice.

### 3.2. ChAT^+^ Synaptic Terminals Are Smaller in Tg2576 than in Wild Type Mice 

In the AD brain, axonal shrinkage is accompanied by synaptic loss in key neurons for memory and cognition. We therefore hypothesized that neuronal shrinkage in skeletal muscle may correlate with synaptic loss, with a particular impact on cholinergic synapses. To identify ACh synaptic terminals, we double stained longitudinal TA sections from Tg2576 and wild type mice with antibodies raised against NFL and choline acetyltransferase (ChAT), the enzyme necessary for ACh synthesis and used as a marker for Cholinergic neurons [26]. ChAT mainly concentrates in the nerve terminals [27], and we consistently detected ChAT signal around synapses, whose identity was confirmed on the basis of their typical morphology (Figure 2A and Appendix A. See also Figure 3A for NFL/ChAT signal).

The total NFL^+^ signal labeling synaptic area was significantly reduced in Tg2576 mice compared to their wild type littermates (t_(8)_ = 4.107, *p* = 0.0034, Figure 2B). This effect is due to a different distribution of the NFL+ area size between genotypes, which reached 1600 μm^2^ in wild type mice and a maximum size of 1200 μm^2^ in Tg2576 mice, as evidenced by the left-shift of Tg2576 vs. wild type cumulative frequencies representing the distribution of these data (Kolmogorn–Smirnoff test: D = 0.2285, *p* = 0.010, Figure 2C). Together with the axonal length values reported in Figure 1C, these data indicate that Tg2576 mice display reduced skeletal muscle innervation and a smaller synaptic area compared to wild type controls.

To assess ChAT expression within synaptic areas, we quantified co-staining for NFL and ChAT from TA sections (Figure 3A). We found that the total overlapping NFL+/ChAT+ area was larger in wild type mice compared to Tg2576 (t_(8)_ = 2.689, *p* = 0.0275, Figure 3B), suggesting that the overall levels of ChAT are reduced in the synapses of Tg2576 mutant mice.

### 3.3. Reduced Expression of Chrna1 in TA Muscle from Tg2576 Mice

To further explore the state of cholinergic synapses, we investigated the expression of nicotinic acetylcholine receptor, subunits α1 and γ (Chrna1 and Chrng), which are commonly expressed in the skeletal muscle [28]. Specifically, we analyzed by qRT-PCR the expression level of Chrna1 and Chrng in TA muscle isolated from six-month-old Tg2576 and wild type mice. The mRNA level of Chrna1 is significantly reduced in Tg2576 mice compared with reference wild type mice (t_(8)_ = 3.183, *p* = 0.0129; Figure 4). Instead, Chrng is not appreciably expressed in TA muscle and is not significantly modulated among genotypes (Appendix A). Collectively, these data suggest that the synaptic alteration revealed in AD mice by imaging analysis of NMJ is also associated with a down-regulation of molecular components of the cholinergic circuit at the transcriptional level.

From the same TA samples, we also analyzed by qRT-PCR the expression levels of other GABAergic, glutamatergic and dopaminergic markers, as these neurotransmitter systems are known to be altered in AD brains. We did not detect significant differences among genotypes in the mRNA expression of GABAergic receptors 1 (Gabbr1), glutamate decarboxylases 1 (Gad1), and dopaminergic receptors 1 (Drd1) even though we noted higher expression levels and variability in AD mice (Appendix A). These data further support the specificity of ACh dysfunction in skeletal muscle of Tg2576 mice.

## 4. Discussion

In Alzheimer’s disease, cognitive decline directly correlates with loss of muscular tone and strength [29]. Cognitive failure depends on degeneration of cholinergic neurons in the brain, which in turn has been associated with the gradual accumulation of Aβ (amyloid beta peptide) and other APP (amyloid precursor protein) products. Deposits of APP products have also been found in the peripheral nervous system [13] and could be involved in the damage of the cholinergic neurons that control muscle function in the peripheral system. Here, we used a mouse model of progressive Aβ accumulation to assess whether cholinergic loss is also evident in their skeletal muscle. Tg2576 mice display Aβ accumulation in the spinal cord [20] and skeletal muscle (Appendix A). Furthermore, these mice have been shown to manifest several symptoms that can be associated with loss of muscle mass and strength including shorter, narrower and slower strides or altered duty cycle and step patterns compared to controls [20,21]. Hence, Tg2576 mice represent a valid model to study sarcopenia in AD.

Our data show that in the Tg2576 mouse model of Aβ over-expression there is a reduced axonal innervation of TA skeletal muscle compared to control wild type mice (Figure 1). This evidence is based on NFL-staining which labels overall axonal processes. Hence, further studies may be needed to better investigate the possible damage of other neuritic processes, including dendrites, and to understand whether the axonal loss is specific for motor neurons or other neuronal subtypes within the skeletal muscle. 

Furthermore, we found that synaptic terminals in these mice are smaller than the ones of wild type littermates (Figure 2) and seem to express less choline acetyltransferase enzyme (Figure 3). These evidences are in line with two previous studies. Seo et al. (2010) [20] reported that Tg2576 mice loose lumbar cord cholinergic neurons, while Monteiro-Cardoso and colleagues [18] found loss of choline acetyltransferase and catalase activities in the skeletal muscle of another mouse model for AD. Together, these data support the possibility of a poor cholinergic innervation of skeletal muscles in AD mice. In this framework, we also provide details concerning neuronal morphology and subcellular identification of cholinergic dysfunction. Furthermore, we report that the expression of Nicotinc Acetylcholine Receptor (nAChRs) subunits α1 (also referred to as Chrna1) is significantly reduced at the mRNA level in Tg2576 mice compared to wild type controls (Figure 4). The link between Aβ toxicity and nAChRs has been extensively detailed in the AD brain [28]. However, nothing is currently known about possible AD-related effects on the nicotinic subunits that function at the muscle. Our data point for the first time to a possible impact of Aβ on the expression of genes encoding for at least one of these subunits.

Altogether, morphological and molecular data reported in this study indicate that cholinergic dysfunction, which has been widely described in the AD brain and includes deficits in nAChRs expression and reduced ChAT activity, is also evident in the skeletal muscle of the Tg2576 AD model. An important question emerging from this new evidence is whether acetylcholinesterase (AChE) inhibitors, that act to enhance cholinergic transmission, can ameliorate sarcopenia symptoms in AD. Currently, very few studies directly investigated motor symptoms in AD patients treated with AChE inhibitors are present [30]. Among these inhibitors, donepezil and galantamine have been shown to improve motor functions such as gait velocity, stride time and praxis [31,32,33], but these effects may depend on increased attention in treated patients. Rather, adverse side effects of AChE inhibitors include fatigue and muscle cramp [34], pointing to a possible exacerbation of sarcopenia symptoms. However, one study showing that donepezil improves muscle atrophy in a mouse model of ischemia [35] paves the way to future investigations on the effect of this and other AChE inhibitors on sarcopenia in Tg2576 mice.

Reduced lean mass and loss of muscle force is a typical sign of early AD stages [3]. Similarly, weight loss, motor impairment, gait disorders and stride difficulties can occur before AD diagnosis and typically correlate with a fast clinical progression [5,6,36]. Hence, a diagnosis of sarcopenia can be an early indication for the AD onset and related biomarkers including the identification of NFL amount from CSF samples [37] or muscle biopsies [4] can become valid tools for the prediction of an AD clinical course.

The synaptic and neuritic loss in the skeletal muscle of Tg2576 mice mimic our previous results collected in the CNS of the same mice. In fact, we previously found that synaptic dysfunction and shrinkage of excitatory neurons are evident in the brain of Tg2576 mice starting from three months of age, in close relation to the onset of cognitive decline, and that Aβ aggregates have a causal role in the damage of brain synapses [24,38]. Since the Tg2576 mouse is a model of gradual Aβ accumulation and that Aβ aggregates have been reported in PNS of AD patients and mouse models, our data supports the hypothesis that an Aβ-driven mechanism is also involved in the morphological damage of skeletal muscle neurons and synapses.

For our investigation we used six-month-old mice, corresponding to an age in which AD pathology is already evident in Tg2576 mice [24,39]. However, other time points may be needed to better understand the progression of cholinergic dysfunction in skeletal muscle and the specific contribution of age to such dysfunction. Furthermore, in support of future studies at different time points, we noted a high variability in the mRNA expression level of GABAergic, glutamatergic and dopaminergic markers in AD mice. This variability may indicate the onset of dysfunctions in these systems, which may exacerbate with age.

Although we did not directly investigate the cellular or molecular mechanisms through which Aβ damages cholinergic neurons in skeletal muscle, we can speculate that events acting in the CNS, including mitochondrial dysfunction or calcium dyshomeostasis [23], could play a pivotal role. Furthermore, there is evidence that in the enteric section of the peripheral nervous system, Aβ oligomers tend to accumulate in cholinergic neurons [40], a fact that could explain the vulnerability of this neuronal class. Even though we only focused on the innervation of the cholinergic system, our study paves the way to further investigations targeting the noradrenergic system in skeletal muscles, in light of recent studies that found sympathetic (noradrenergic) innervation of skeletal muscles [41].

It should also be taken into account that peripheral mechanisms can act in parallel with other central factors that explain sarcopenia, including loss of dopaminergic regulation of the motivational system or alterations in executive functions [42] which have been associated with AD progression.

## 5. Conclusions

Our results sustain the hypothesis that loss of cholinergic innervation in the skeletal muscle of AD mice could be responsible for muscle weakening and sarcopenia. Although further studies are necessary to dissect the molecular factors linking APP processing to cholinergic loss and possibly muscle dysfunction (e.g., sarcopenia), our data raise the interesting possibility that targeting the cholinergic system could be useful for the treatment of sarcopenia and other muscle pathologies that accompany AD clinical progression.

## Figures and Tables

**Figure 1 brainsci-11-01245-f001:**
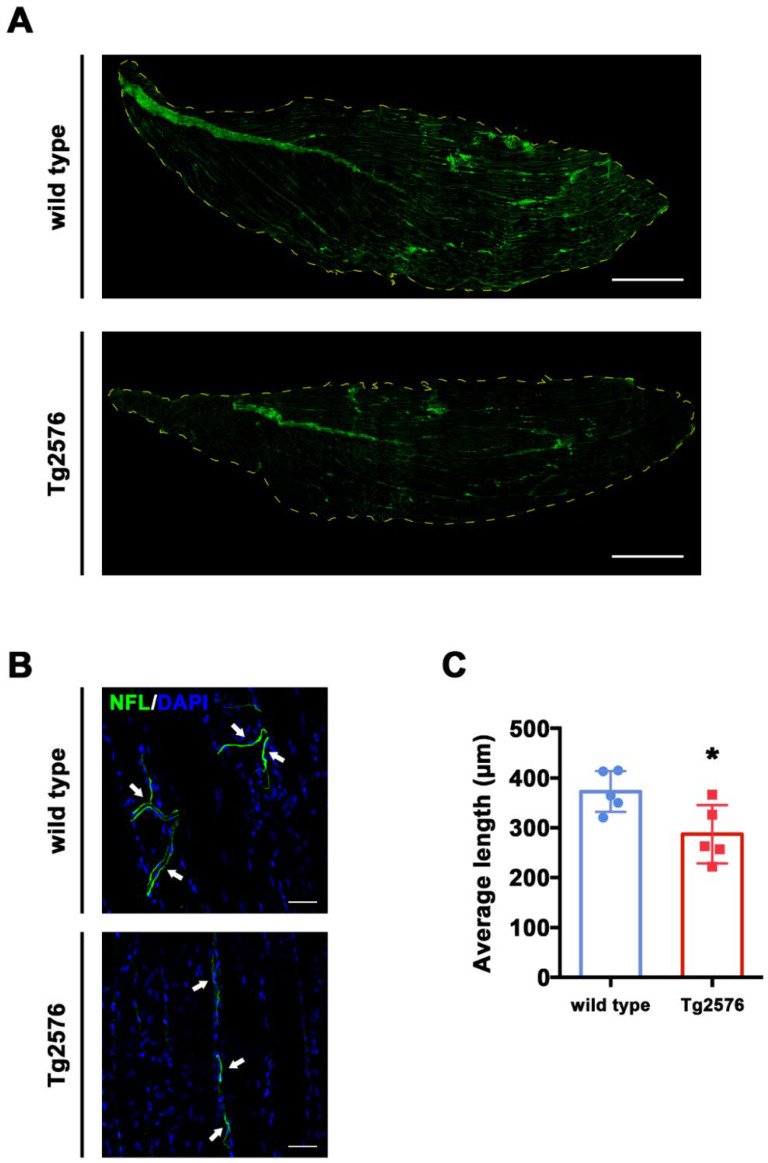
Tg2576 mice display reduced neuritic length in TA. (**A**) Representative images of whole longitudinal section of TA muscle of wild type and Tg2576 mice. Green stain corresponds to NFL positive area. Dotted yellow line corresponds to TA edges. Scale bar = 1 mm (**B**) Representative images of NFL-stained TA sections of Tg2576 and wild type mice. Sections were counterstained with DAPI in order to detect cell nuclei. White arrows point to neurofilaments. Scale bar = 50 µm. (**C**) Histogram reporting average axonal length/ROI ± SEM from NFL-stained TA slices of Tg2576 and wild type mice (*n* = 5 mice per genotype). Unpaired *t* test was used for comparison (* = *p* < 0.05).

**Figure 2 brainsci-11-01245-f002:**
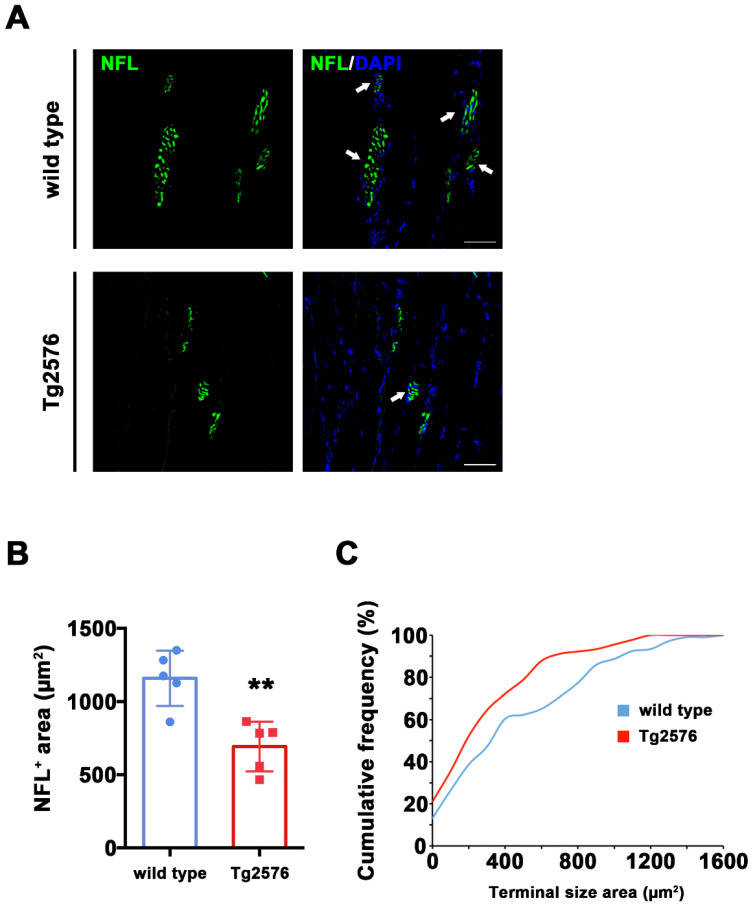
Reduced pre-synaptic area in TA muscle of Tg2576 mice. (**A**) Representative images of TA sections, stained with antibodies anti-NFL (green) from Tg2576 and wild type mice. Sections were counterstained with DAPI in order to detect cell nuclei. White arrows point to pre-synaptic area. Scale bar = 50 µm. (**B**) Histogram reporting the average area of NFL signal in pre-synaptic areas from TA sections of Tg2576 and wild type mice (*n* = 5 mice per genotype). Unpaired *t* test was used for comparison (** = *p* < 0.01). (**C**) Graph of cumulative frequencies reporting the distribution of NFL-positive signal sizes of both genotypes (*n* = 5 mice per genotype). Kolmogorov–Smirnov test was used for comparisons (*p* < 0.05).

**Figure 3 brainsci-11-01245-f003:**
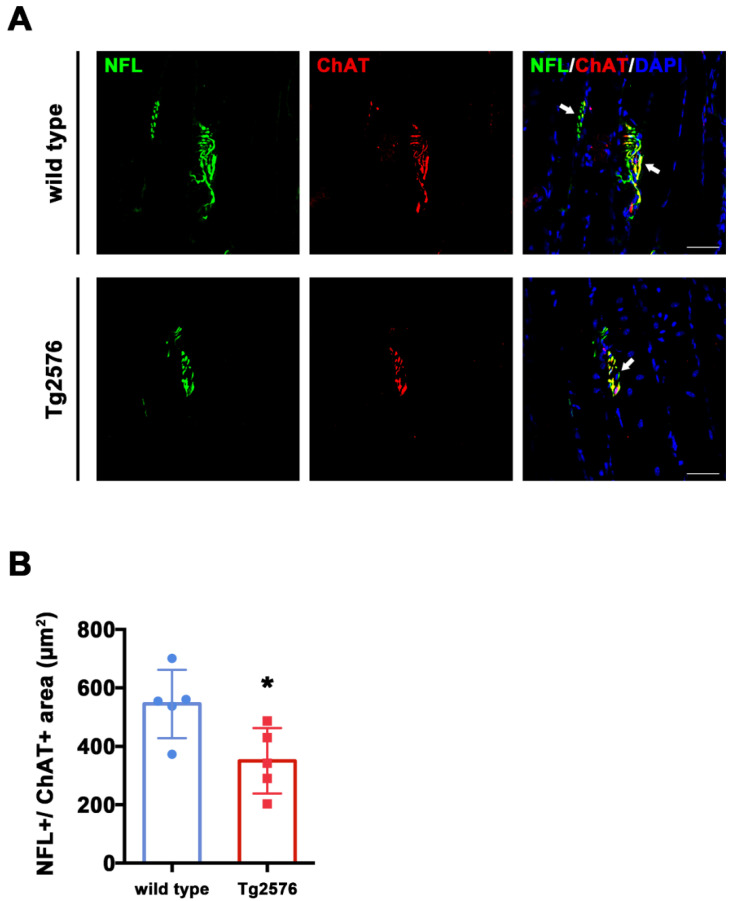
Tg2576 mice exhibit reduced ChAT^+^ area in pre-synaptic terminals of TA muscles. (**A**) Representative images of TA sections, stained with antibodies anti-NFL (green) and anti-ChAT (red), from Tg2576 and wild type mice. Sections were counterstained with DAPI in order to detect cell nuclei. White arrows point to pre-synaptic area, positive for both markers. Scale bar = 50 µm. (**B**) Histogram reporting the average area of ChAT signal in pre-synaptic surface, identified by the presence of both NFL and ChAT signals, in TA sections of Tg2576 and wild type mice (*n* = 5 mice per genotype). Unpaired *t* test was used for comparison (* = *p* < 0.05).

**Figure 4 brainsci-11-01245-f004:**
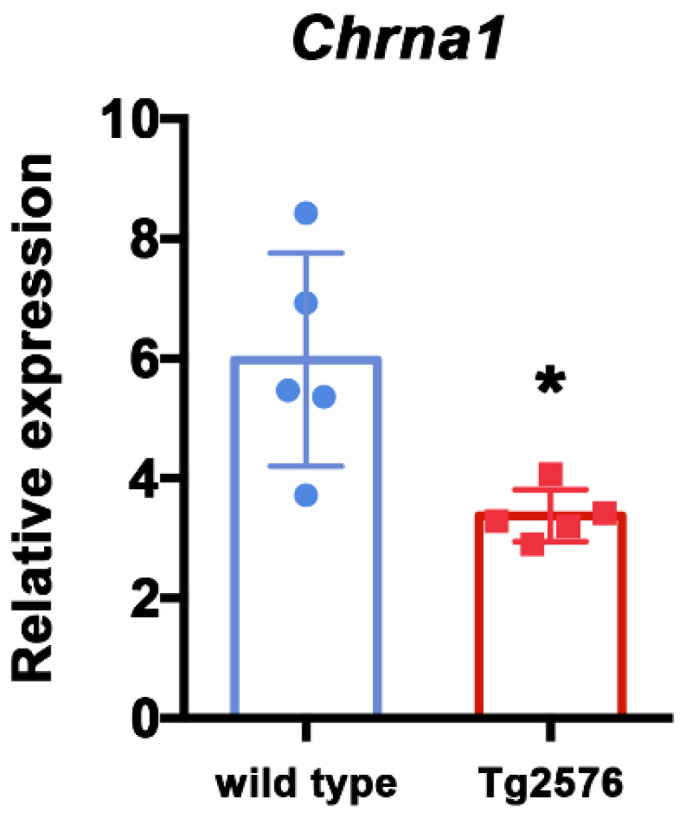
Cholinergic receptor nicotinic alpha polypeptide 1 (Chrna1) is down-regulated in skeletal muscle of Tg2576 mice. Expression analysis of Chrna1, by qRT-PCR, on TA muscles derived from 6-month-old female wild type and Tg2576 mice. Data are reported as relative to housekeeping gene TBP, and represented as mean ± SEM (*n* = 5 mice); unpaired *t* test was used for comparison (* = *p* < 0.05).

## Data Availability

Data supporting reported results will be available at Institute’s repository page and provided on request.

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
