# Peer review of "Peripheral Nerve Impairment in a Mouse Model of Alzheimer’s Disease"

_brainsci, 2021, doi:10.3390/brainsci11091245_

Round 1

Reviewer 1 Report

The current manuscript describes the loss of cholinergic innervation in the skeletal muscle of AD mice could be responsible for muscle weakening and sarcopenia. Their data showed using a Tg2576 mouse model of Aβ over-expression, there is a reduced axonal innervation of tibials anterior skeletal muscle compared to control wild type mice. This evidence is based on NFL-staining that labels overall axonal processes. This is a well written manuscript and needs to address the following concerns.  

  1. Spell check is needed.
  2. 2 and Fig. 3 are same. Please explain the reason for including two.
  3. How other markers such as Dopamine, and GABA are affected in this mouse model since their levels are altered in the AD brain too.
  4. Does using any AChE inhibitors will benefit over the sarcopenia in this model?
  5. Provide the Aβ staining picture for both control and Tg2576 mice as a proof-of-principle.

Author Response

We thank the reviewer for comments that have been taken in consideration in the new version of the manuscript. In this new version: 

We check for type errors and correct them. We apologize for these mistakes. 

We simplified Figure 2 in order to avoid redundancies. 

As suggested, we measured the mRNA expression of other GABAergic, Glutamatergic and Dopaminergic markers that are commonly altered in the AD brain. These data are now included in section 3.3.  

Unfortunately,  due to unavailability of Tg2576 mice and Institutional review board approval, we could not run a study testing the effect of AChE inhibitors. Yet, we added in the manuscript a detailed discussion that refers to important studies investigating the impact of  AChE  inhibitors on motor functions and muscle strength in humans, as well as one study that investigates the effects of such treatment in mice, and that potentially paves the way to future studies in Tg2576 mice. 

As suggested, we provide an Aβ staining picture for both control and Tg2576 mice as a proof-of-principle (Supplementary Figure 1). For this, we used an antibody that is raised against Aβ fragments and recognizes human APP that is over-expressed in Tg2576 mice. 

Reviewer 2 Report

Torcinaro et al. studied the peripheral axonal loss in a genetic model for Aβ accumulation (Tg2576 mice). Authors demonstrated in the AD model a reduced neuritic length with loss of cholinergic innervation and nAChRs. Authors suggested that Ab accumulation determine degeneration of skeletal muscle neurons and synapses, as similarly occurs in CNS. The manuscript is well-written and easy reading. I have just a couple of comment:

- Did authors study motor performance in the mice (Tg2576 and wt) in order to compare it with pathological results (neuritic length, cholinergic innervation, Ach receptors)?

- Figure 2A and 3A represent the same results. Authors can choose one of them, avoiding data redundance.

- Specify in method section (point 2.2) the timing of mice euthanasia (6 months for both species).

- Some typos are present, like “dendtites” and “subtipes” in the discussion section.

Author Response

We thank Reviewer 2 for helpful comments. 

As mentioned by the reviewer, measuring motor performance in mouse model would had be useful to correlate motor symptoms with pathological events. Unfortunately, we don not have Tg2576 mice and Institutional review board available at this moment. Still, we can infer the sarcopenia-like symptoms from data available in the literature about these mice. At least two papers have been shown that Tg2576 mice display typical signs of muscle fatigue, including short, narrow and slow strides or altered duty cycle. These studies (See et al., 2010; Nyul-Toth et al., 2021) are now mentioned in both the introduction and the discussion sections of the new manuscript version. 

As suggested from reviewer, we modified Figure 2A in order to avoid data redundance.

Timing for mice euthanasia has been added in section 2.2.

Type errors have been corrected. We apologize for these misspellings. 

Round 2

Reviewer 1 Report

The authors addressed my points. 

However, I was not able to open the supplementary file to go through the information provided such as Supplementary Figures 1 and 2

Author Response

Please, find the supplementary material in pages 17-19 of the present file
